# Reconstructing Reliable Powder Patterns from Spikelets (Q)CPMG NMR Spectra: Simplification of UWNMR Crystallography Analysis

**DOI:** 10.3390/molecules26196051

**Published:** 2021-10-06

**Authors:** Andrii Mahun, Sabina Abbrent, Jiri Czernek, Jan Rohlicek, Hana Macková, Weihua Ning, Rafał Konefał, Jiří Brus, Libor Kobera

**Affiliations:** 1Institute of Macromolecular Chemistry of the Czech Academy of Sciences, Heyrovskeho nam. 2, 162 06 Prague 6, Czech Republic; mahun@imc.cas.cz (A.M.); abbrent@imc.cas.cz (S.A.); czernek@imc.cas.cz (J.C.); mackova@imc.cas.cz (H.M.); konefal@imc.cas.cz (R.K.); brus@imc.cas.cz (J.B.); 2Department of Physical and Macromolecular Chemistry, Faculty of Science, Charles University, Hlavova 8, 128 40 Prague 2, Czech Republic; 3Department of Structural Analysis, Institute of Physics of the Czech Academy of Sciences, Na Slovance 2, 182 21 Prague 8, Czech Republic; rohlicek@fzu.cz; 4Department of Physics, Chemistry and Biology (IFM), Linköping University, SE-581 83 Linköping, Sweden; weihua.ning@liu.se

**Keywords:** UWNMR, spikelets NMR spectra, NMR software, spectral envelope

## Abstract

Spikelets NMR spectra are very popular as they enable the shortening of experimental time and give the possibility to obtain required NMR parameters for nuclei with ultrawide NMR patterns. Unfortunately, these resulted ssNMR spectra cannot be fitted directly in common software. For this reason, we developed UWNMRSpectralShape (USS) software which transforms spikelets NMR patterns into single continuous lines. Subsequently, these reconstructed spectral envelopes of the (Q)CPMG spikelets patterns can be loaded into common NMR software and automatically fitted, independently of experimental settings. This allows the quadrupole and chemical shift parameters to be accurately determined. Moreover, it makes fitting of spikelets NMR spectra exact, fast and straightforward.

## 1. Introduction

In the last fifty years, NMR spectroscopy has undergone progressive development. One of the first NMR publications sixty years ago opened the door for NMR studies, providing valuable information about the molecular structure and dynamics in solid systems [1]. Almost simultaneously, the classic paper “*Nuclear Electric Quadrupolar Interactions in Crystals*” was published by R. V. Pound [2], and, over the next few years, many papers dealing with the analysis of quadrupolar NMR powder lineshapes were published. Since ca. 75% of NMR active nuclei have a quadrupolar character (spin *I* > ^1^/_2_) and, therefore, provide very broad and complicated spectral lines, this area is still under scrutiny. Many of these quadrupolar nuclei are unreceptive to the traditional NMR experiments due to their low natural abundance or low gyromagnetic ratio. Moreover, additional complications arise from the anisotropic broadening of powder patterns which dramatically reduce the signal-to-noise ratio (SNR). As the resulting powder patterns can reach from tens kHz to a few MHz in width, they cannot be acquired using techniques such as the single-pulsed (SP) experiment because of the inability to efficiently excite the whole powder pattern, signal loss during the ring-down of the probe and the recovery of the receiver immediately after the pulse. The description of dominant anisotropic NMR interactions is necessary for the understanding of quadrupolar nuclei behavior such as (i) the quadrupolar interaction [3], (ii) chemical shift/chemical shift anisotropy [4] or (iii) anisotropic broadening induced by the presence of unpaired electrons [5]. Residual interactions, such as dipolar and scalar interactions, can be, in this case, omitted due to their relatively low values.

Quadrupolar interaction exists between local electric field gradients (EFGs) and the nuclear quadrupolar moment (*Q*). The traceless and symmetric local EFG tensor is defined as |*V_33_*| ≥ |*V_22_*| ≥ |*V_11_*|. The quadrupolar coupling constant—*C_Q_*—and the asymmetry parameter—*η_Q_*—are derived from the principal values of the EFG tensor (*V_33_* = *eq*) as follows: *C_Q_* = *e^2^qQ/h* and *η_Q_* = (*V_11_* − *V_22_*)/*V_33_*. These two parameters define the quadrupolar interaction and can be extracted from recorded NMR powder patterns. They allow an insight into spherical and axial symmetry of the local electronic environment. In recent decades, many experimental techniques have been developed enabling the acquirement of quadrupolar parameters (i.e., DOR [6], DAS [7], STMAS [8,9], MQMAS [10]). However, most of these techniques can only be used for patterns narrower than 150 kHz in breadth. For wider spectral lines, usually defined as Ultra-Wideline NMR patterns [11,12,13] (UWNMR), special NMR techniques based on piecewise spectral acquisition [14,15,16] or spin echo [17] have to be applied. The term “piecewise technique” is used when the full pattern is obtained by conducting and co-adding a series of experiments where each provides a section of individual points or sub-spectra. This technique uses two possible approaches: (i) field stepping, where the transmitter frequency is constant and the magnetic field is incremented or (ii) frequency stepping, where the transmitter frequency is incremented at a constant magnetic field. While these techniques provide valuable results and are used to this day, the weaker spectral resolution and extensive experimental time represent their undismissible disadvantage.

On the other hand, “spin echo techniques”, originally proposed by Hahn [17] and composed of two pulses with an inserted delay (90°-*τ*-180°-*aq*.), are designed for the effective refocusing of dephased spins and to provide spectral lines with a sufficient resolution. A modified technique known as quadrupolar or solid echo, commonly used for measurements of quadrupolar nuclei, uses shorter pulses, (i.e., 90°-*τ*-90°-*aq*) [18,19]. In the case of UWNMR, the spin echo approach can be used in combination with the frequency stepping technique and called “variable-offset cumulative spectroscopy” (VOCS) [16]. Furthermore, the “Hahn echo” approach was further extended by Carr and Purcell for measurements of T_2_ relaxation times (the second delay was spaced between the second pulse and acquisition time 90°-*τ*-180°-*τ*-*aq*.) [20]. Subsequently, Meiboom and Gill modified the Carr–Purcell sequence, adding a train of 180° pulses with a 90° phase shift with respect to the first pulse (90°-*τ*-(180°-*τ*-*aq*.)_n_) [21]. The resulting Carr–Purcell–Meiboom–Gill (CPMG) sequence is a suitable tool for the enhancement of SNR of solid-state NMR (ssNMR) spectra. Later, pioneering work was presented by Sinfelt for nuclei with spin *I* = ^1^/_2_ [22]. Twenty years ago, Larsen used the CPMG sequence for quadrupolar nuclei (QCPMG) [23] and a further combination with the frequency-stepped experiment allowed the recording of UWNMR spectra with satisfactory SNR [24]. Then, each echo-train was recorded as a single FID and was Fourier-transformed as an entity and then combined into a “spikelets” spectrum. However, obtaining UWNMR spectra in one piece was still a challenge and, therefore, an alternative approach was developed which applies constant transmitter frequencies with the modulation of either pulse amplitude or phase in order to excite a broad enough frequency range. In 2007, Bhattacharyya and Frydman replaced the regular pulses in the Hahn echo with an adiabatic sequence [25] termed “wide-band, uniform-rate, and smooth truncation” (WURST) introduced by Kupce and Freeman in 1995 for the uniform excitation of a broad frequency range, [26] resulting in the so-called WURST-echo sequence [27]. Subsequently, Schurko et al. published the WURST-QCPMG sequence where all pulses were substituted by WURST-80 pulses [28]. This approach allows acquiring UWNMR spectra in one piece that would normally require frequency-stepped (VOCS) techniques. Finally, Schurko et al. successfully applied this approach to various spin-^1^/_2_ nuclei (i.e., ^119^Sn, ^195^Pt, ^199^Hg and ^207^Pb), which are known to exhibit strong CSA interactions, making it difficult to obtain the powder pattern by frequently used techniques (SP/MAS, MQ/MAS, etc.) [29].

Usually, required NMR parameters (quadrupolar parameters (*C_Q_*, *η_Q_*) and chemical shift anisotropy (CSA), etc.) are extracted from resulting (Q)CPMG spectra using a careful manual simulation and fitting of the powder pattern. This approach is relatively time-consuming and should be supported by data obtained from quantum chemical calculations. Here, another complication arises as an agreement between calculated data and experimental data is not always guaranteed.

Therefore, we developed a tool for the easy extraction of the spectral envelopes of the obtained spikelets powder pattern from (Q)CPMG NMR spectra, which can be used for the automatic fitting procedure. The outcome of this automated procedure is an accurate spectral envelope of the UWNMR pattern that can be easily used for the extraction of valuable NMR parameters. The developed software was verified using materials with a known structure and NMR parameters. Additionally, the developed approach was compared with the existing method for obtaining spectral envelopes by (time-domain) the co-addition of individual spin echoes followed by a Fourier transformation (FFT). As a final step, unknown NMR parameters were successfully extracted from reconstructed newly obtained single continuous spectral lines and confirmed by DFT calculations.

## 2. Results and Discussion

### 2.1. Verification of the USS Software Using Model Spectra

For the validation and verification of reconstructed spectral envelopes obtained using USS software, a diverse group of NMR active nuclei was selected. Both quadrupolar (^137^Ba, ^35^Cl and ^27^Al) and spin-^1^/_2_ nuclei (^207^Pb and ^199^Hg) were chosen and experimental WURST-(Q)CPMG NMR spikelets spectra of Ba(NO_3_)_2_, BaCO_3_, BaCl_2_∙2H_2_O, AlMes_3_ + AlMes_3_∙THF, Pb(HCOO)_2_ and Hg(CH_3_COO)_2_, with known structures and NMR parameters were acquired (see Figure 1, expt.). These model compounds possess various structural features, allowing us to demonstrate wide possibilities of this approach. These chosen features were as follows: (i) quadrupolar nuclei with one crystallographic site (Ba(NO_3_)_2_ and BaCO_3_) but with different quadrupolar coupling constants; (ii) quadrupolar nuclei with two crystallographically different sites (BaCl_2_∙2H_2_O) of one investigated nucleus; (iii) quadrupolar nuclei containing two distinct crystalline compounds (AlMes_3_ + AlMes_3_∙THF); (iv) spin-^1^/_2_ nuclei with the dominant contribution of CSA (Pb(HCOO)_2_ and Hg(CH_3_COO)_2_).

The set of obtained spectra was successfully processed by the USS software, and the final spectral lines were obtained (Figure 1, envelope). In the case of the ^137^Ba WURST-QCPMG NMR spectrum of BaCO_3_, artifact interferences were observed (Figure 1b; see full spectrum Appendix A). Similarly, in the ^27^Al WURST-QCPMG NMR spectrum of the AlMes_3_ + AlMes_3_∙THF sample, the impurities gave additional signals (Figure 1d). Therefore, the structure of BaCO_3_ was also confirmed by the XRPD analysis (Appendix A), and a corresponding XRPD pattern of AlMes_3_ + AlMes_3_∙THF is described in [30].

As the next step, these spectra were transferred to two commonly used programs for NMR data processing (Bruker TopSpin and DMfit). The fitting of the spectral lines was easily accomplished using an automatic procedure and corresponding NMR parameters were obtained. These fittings of spectral lines (Figure 1, TS and DMfit) were in good agreement with the literature data (Figure 1, lit. data). The comparison of experimental and literature-based NMR parameters is listed in Table 1, showing only small differences within the experimental error.

It has been shown [11,31] that the envelope of (Q)CPMG spikelets spectra can be obtained by the summation of individual spin echo FIDs and the subsequent Fourier transformation. The envelopes (of selected compounds) obtained by the co-addition of Fourier-transformed spin echo FIDs were compared with envelopes reconstructed from the USS software, see Figure 2. Clearly, both approaches provided basically the same spectral profiles and the fitted NMR parameters (Appendix A) were in great consistency. However, as the USS software built the spectral envelopes from the maximum points (maxima were also taken from the noise regions), this resulted in a higher SNR, Figure 2, caused by the magnitude processing (‘mc’ command in TopSpin software) of the spikelets NMR spectra. Although spectral envelopes can be obtained by the above-mentioned spin echo co-addition method, the developed approach (using USS software) was both (i) time-saving and (ii) easy to use. Obtaining the spectral profile by the summation of spin echo requires complicated manipulation with raw data (separation of the initial FID into a set of individual spin echoes with a subsequent FTT procedure of each and manual co-addition of the single spectra into the sum envelope). On the other hand, the USS software provided a fast and intuitive way to generate a spectral envelope without any manipulations with raw data (only a few clicks after loading a spikelets spectrum were needed to obtain the spectral envelope). The developed approach was especially beneficial in the case of UWNMR spikelets spectra obtained by the VOCS technique, since such spectra comprise of tens, in extreme cases hundreds, of sub-spectra.

**Table 1 molecules-26-06051-t001:** Comparison of experimental NMR parameters with literature data for chosen model compounds.

Compound	Investigated Nuclei	*δ_iso_* (ppm)	*C_Q_* (MHz)	*η_Q_*	*Ω* (ppm)	*κ*	*α* (deg)	*β* (deg)	*γ* (deg)	Ref.
Ba (NO_3_)_2_	^137^Ba	*−42 (8)*	*7.0 (0.10)*	*0.01 (0.01)*	*25 (20)*	*0.80 (0.20)*	*40 (20)*	*10 (25)*	---	[32]
		−49 (5)	6.9 (0.10)	0.02 (0.01)	26 (15)	0.80 (0.20)	43 (15)	11 (20)	---	this work ^1^
		−47 (5)	6.9 (0.10)	0.02 (0.01)	24 (15)	0.70 (0.20)	53 (20)	32 (20)	---	this work ^2^
BaCO_3_	^137^Ba	*50 (200)*	*17.4 (0.60)*	*0.33 (0.04)*	*150 (150)*	*0.50 (0.50)*	---	---	---	[32]
		76 (150)	16.8 (0.50)	0.34 (0.03)	149 (130)	0.50 (0.40)	---	---	---	this work ^1^
		108 (150)	16.6 (0.50)	0.34 (0.03)	183 (140)	0.40 (0.40)	---	---	---	this work ^2^
BaCl_2_∙2H_2_O	^35^Cl (site A)	*163 (2)*	*2.19 (0.08)*	*0*	*50 (25)*	*−0.80 (0.20)*	*85 (20)*	*32 (10)*	*60 (20)*	[33]
		167 (10)	2.27 (0.12)	0	39 (25)	−0.60 (0.20)	89 (20)	47 (15)	49 (20)	This work ^1^
		164 (10)	2.20 (0.12)	0	43 (25)	−0.60 (0.20)	100 (25)	30 (15)	66 (20)	This work ^2^
	^35^Cl (site B)	*156 (2)*	*3.42 (0.08)*	*0.30 (0.10)*	*50 (25)*	*0.20 (0.25)*	*20 (15)*	*8 (10)*	*0 (20)*	[33]
		154 (10)	3.48 (0.15)	0.30 (0.10)	52 (25)	0.20 (0.20)	22 (15)	12 (10)	0 (25)	This work ^1^
		158 (10)	3.49 (0.15)	0.30 (0.10)	47 (25)	0.10 (0.20)	15 (15)	15 (10)	5 (25)	This work ^2^
AlMes_3_	^27^Al (1)	*240 (20)*	*49.2 (0.50)*	*0.01 (0.01)*	*126 (10)*	*−0.99 (0.20)*	*180 (20)*	*0 (10)*	*100 (10)*	[30,34]
		242 (15)	49.2 (0.50)	0.01 (*0.01*)	126 (10)	−0.99 (0.20)	170 (15)	0 (10)	110 (10)	this work ^1^
		226 (15)	48.8 (0.50)	0.01 (*0.01*)	118 (10)	−0.99 (0.20)	180 (15)	0 (10)	100 (10)	this work ^2^
AlMes_3_∙THF	^27^Al (2)	*120 (10)*	*27.3 (0.30)*	*0.13 (0.10)*	*46 (5)*	*0.05 (0.10)*	*205 (10)*	*80 (5)*	*200 (10)*	[30]
		130 (10)	27.0 (0.30)	0.17 (0.08)	47 (5)	0.05 (0.08)	195 (10)	83 (5)	210 (10)	this work ^1^
		133 (10)	26.8 (0.30)	0.17 (0.08)	51 (5)	0.07 (0.08)	200 (10)	80 (5)	215 (15)	this work ^2^
Pb (HCOO)_2_	^207^Pb	*−2567 (20)*	*---*	*---*	*720 (20)*	*0.75 (0.10)*	---	---	---	[35]
		*−*2562 (15)	---	---	708 (15)	0.74 (0.10)	---	---	---	this work ^1^
		*−*2562 (15)	---	---	701 (20)	0.75 (0.10)	---	---	---	this work ^2^
Hg (CH_3_COO)_2_	^199^Hg	*-2513 (34)*	---	---	*1810 (68)*	*0.89 (0.07)*	---	---	---	[29]
		*−*2485 (40)	---	---	1814 (65)	0.82 (0.07)	---	---	---	this work ^1^
		*−*2488 (40)	---	---	1811 (60)	0.82 (0.07)	---	---	---	this work ^2^

*δ_iso_* = *(δ_11_* + *δ_22_* + *δ_33_*)/3, *C_Q_* = *eQV_33_*/*h*, *η_Q_* = *(V_11_*–*V_22_*)/*V_33_*, *Ω* = *δ_11_*_–_*δ_33_*, *κ* = 3 (*δ_22_*-*δ_iso_*)/*Ω*; Euler angles are defined in [36]; ^1^ NMR parameters obtained using Bruker TopSpin software; ^2^ NMR parameters obtained using DMfit software.

### 2.2. Investigation of Compounds with Unknown NMR Parameters

As the developed UWNMRSpectralShape software was validated on compounds with known NMR parameters, the next step was to use it for extracting unknown NMR parameters. Experimental ^71^Ga, ^209^Bi and ^115^In WURST-QCPMG NMR spikelets spectra (Figure 3, expt.) of compounds with unknown NMR parameters were recorded and continuous lines obtained using the developed USS software. Subsequently, the spectral envelopes with a small *C_Q_* value were fitted only in the TopSpin software, while the spectral envelopes with a dominant *C_Q_* contribution were fitted in the TopSpin as well as in DMfit software.

Initially, a relatively simple ^209^Bi NMR spectrum of the Cs_2_AgBiBr_6_ sample was examined. The profile of the experimental WURST-QCPMG NMR spikelets spectrum exhibited a symmetric lineshape with only one signal (Figure 3a and Appendix A). X-ray crystallographic data of the Cs_2_AgBiBr_6_ system (Appendix A) indicated a high symmetry (cubic) atomic arrangement around the Bi nucleus, as anticipated on the basis of the defined Fm-3m space group [37]. Simultaneously, the ^209^Bi nucleus is known to yield broad signals due to the considerable quadrupolar moment (*eQ*) and small values of the *T_2_* relaxation time constant, even in symmetric ^209^Bi site environments [38,39,40]. The combined effect of cubic symmetry and quadrupolar moment of Bi nuclei resulted in a relatively broad symmetric spectral lineshape. NMR parameters fitted from the reconstructed lineshape are listed in Table 2. (The spectral envelope was fitted with a small *C_Q_* value and a relatively large value of the line-broadening parameter, see Table 2 and Figure 3a.)

Somewhat more complex, the ^71^Ga WURST-QCPMG NMR spikelets spectrum of the Ga(NO_3_)_3_∙9H_2_O sample showed two distinct peaks, which indicated two gallium sites with different local environments (Figure 3b and Appendix A). A subsequent XRPD analysis of the sample (Appendix A) confirmed the presence of two phases, namely, gallium nitrate hydrate (Ga(NO_3_)_3_∙9H_2_O) and gallium nitrate hydrate hydroxide (Ga_13_(NO_3_)_15_(OH)_24_∙24H_2_O). The presence of two distinct crystalline hydrates was the consequence of strict sample handling under an inert atmosphere (Ar). In laboratory conditions, only gallium nitrate hydrate (Ga (NO_3_)_3_∙9H_2_O) was present. A relatively narrow and symmetric lineshape of the resulting ^71^Ga NMR signals suggested small values of the quadrupolar coupling constant (*C_Q_*) and CSA parameters. Thus, the ^71^Ga NMR signals were fitted accordingly—using Lorentzian/Gaussian approximation (Table 2, Figure 3b). Moreover, the quantification of individual components was possible by the integration of the individual peaks as the resulting spectrum contained well-resolved signals. In this case, the molar ratio of 77% of (Ga (NO_3_)_3_∙9H_2_O) and 23% of (Ga_13_(NO_3_)_15_(OH)_24_∙24H_2_O) was found and confirmed by XRPD data.

As a next step, a system containing nuclei with a dominant quadrupolar contribution was selected. Here, the ^115^In isotope was a suitable candidate due to its adequate sensitivity, gyromagnetic ratio and relatively easily resolved structural features, typical for all triel/icosagen elements. The experimental ^115^In WURST-QCPMG NMR spectrum of In (NO_3_)_3_∙5H_2_O is depicted in Figure 3c and Appendix A. Clearly, the baseline of the experimentally obtained spikelets ^115^In NMR spectrum was distorted due to the probe ringing. In order to remove this distortion, some part of the FID was cut (left-shifted); however, this resulted in a loss of intensity of the right shoulder of the signal (see Appendix A). Therefore, the initial spectrum with a full FID was used for further manipulation, and the option for the correction of the background (noise region) in the USS software was applied. A consequential spectral profile with an even background without any loss of intensity of the main signal was obtained (Figure 3c, envelope). Further, the NMR parameters (Table 2) were fitted using the Bruker TopSpin and DMfit software (Figure 3c, TS and DMfit). Additionally, the crystal structure of the investigated sample was confirmed by an X-ray diffraction phase analysis, verifying the presence of one pure phase (Appendix A).

Finally, the experimental ^71^Ga WURST-QCPMG NMR spikelets spectrum of the [Cl(Me)Ga(O^t^Bu)]_2_ sample (Figure 3d, expt.) was acquired using the WURST-QCPMG experiment combined with the VOCS approach [16]. In order to obtain the whole powder pattern, seventeen sub-spectra had to be co-added. The resulting UWNMR spectrum was exceptionally wide, with ∼2.5 MHz in breadth. Before transferring the obtained spectral profile to the conventional software for fitting, the envelope was slightly smoothed (using the respective option in the USS software); see Figure 3d, envelope. Further, the spectral envelope was fitted as one ^71^Ga site, in both the Bruker TopSpin and DMfit software (Figure 3d, TS and DMfit), and corresponding NMR parameters were obtained.

Experimental NMR parameters for all investigated samples (Ga (NO_3_)_3_∙9H_2_O, Cs_2_AgBiBr_6_, In (NO_3_)_3_∙5H_2_O, and [Cl (Me)Ga(O^t^Bu)]_2_) fitted with the aid of the USS software were compared with theoretical NMR values obtained from the periodic DFT calculations; see Table 2. The theoretical and experimental data showed good agreement in all cases. Furthermore, the corresponding XRPD patterns confirming the presence of individual compounds are listed in Appendix A. It follows that the usability of the presented USS software for the fast and easy extraction of required NMR parameters was successfully verified. 

## 3. Materials and Methods

### 3.1. Chemicals

Barium nitrate—Ba (NO_3_)_2_—, barium carbonate—BaCO_3_—, barium chloride dihydrate—BaCl_2_∙2H_2_O—, mercury (II) acetate—Hg (CH_3_COO)_2_—, gallium nitrate hydrate—Ga(NO_3_)_3_∙9H_2_O—and indium nitrate hydrate—In (NO_3_)_3_∙5H_2_O—were purchased from Sigma-Aldrich and used as received. Trimesitylaluminum—AlMes_3_—and Trimesityl-tetrahydrofuran-aluminum—AlMes_3_∙THF—were prepared and handled as described in [30]. Lead diformate—Pb(HCOO)_2_—was obtained according to the description provided in [35]. Cesium silver(I) bismuth(III) bromide—Cs_2_AgBiBr_6_—was synthetized by the procedure described in the literature data [41]. Bis(*μ*_2_-*t*-butanolato)-dichloro-dimethyl-di-gallium—[Cl(Me)Ga(O^t^Bu)]_2_ (CAJXOC in CCDC database)—was prepared from the following chemicals: lithium bis(trimethylsilyl)amide solution in toluene (0.5 M), anhydrous toluene, and *t*-butanol, bought from Sigma-Aldrich, (St. Louis, MO, USA), gallium trichloride, ultra-dry, from Alfa Aesar (Haverhill, MA, USA). Argon 7.0 (Linde Gas; Prague, Czech Republic) was used as received. All procedures were conducted under argon atmosphere using Schlenk line techniques or glovebox. To prepare [Cl (Me) Ga (O^t^Bu)]_2_, we modified the two-step procedure described by Carmalt et al. [42]. Lithium bis(trimethylsilyl)amide solution (2.2 g, 25 ml, 0.5 M in toluene) was added through septum to suspension of GaCl_3_ (1.1g GaCl_3_, 15 ml toluene) cooled at −78 °C under argon atmosphere. The temperature was slowly allowed to rise and the mixture was refluxed overnight. Toluene was removed by argon flow, and residuum was vacuum-distilled. Crude white liquid [Me (Cl) GaN (SiMe_3_)_2_]_2_ was used in further steps without analysis. [Me (Cl) GaN (SiMe_3_)_2_]_2_ (1 g) was dissolved in toluene (15 ml) and *t*-butanol (3.6 mmol, 340 µL), refluxed overnight, then concentrated under vacuum. Resulting orange residuum was slowly sublimated using oil pump (10^−2^ torr) and heating mantel to obtain colorless crystals of [Cl (Me) Ga (O^t^Bu)]_2_. All prepared samples were stored in a glovebox with inert atmosphere (Ar).

### 3.2. Solid-State NMR

The ssNMR spectra were recorded at 11.7 T using a Bruker AVANCE III HD spectrometer. The 4 mm cross-polarization magic angle spinning (CP/MAS) probe was used. All spectra were recorded at static conditions using WURST-(Q)CPMG technique. In all measurements, 50 μs selective pulse was applied, and high-power ^1^H decoupling (CW) was used to eliminate heteronuclear dipolar couplings. Dried samples were packed into ZrO_2_ rotors and, subsequently, stored at room temperature in an inert atmosphere. All NMR spectra were processed using the Top Spin 3.5 pl2 software package, and spectral lines were simulated using the DMfit 2019 software [43]. Detailed experimental parameters and used external standards are listed in the Appendix A (Appendix A, respectively). The spectra profiles based on the literature data (Figure 1, lit. data) were simulated in Bruker TopSpin software as fitting curves which corresponded to NMR parameters presented in respective scientific articles (see Table 1, [29,30,32,33,34,35]).

### 3.3. X-ray Powder Diffraction (XRPD)

Powder diffraction data were collected using the Bragg–Brentano reflection configuration on the powder diffractometer Empyrean of Panalytical (λCu, K_α_ = 1.54184 Å) that was equipped by a PIXcel3D detector. The phase analysis was performed in the Program HighScore connected with PDF 4+ and CSD databases. Crystallographic data of used samples are summarized in Appendix A.

### 3.4. The Periodic Density Functional Theory (DFT) Calculations

The computational procedure was followed, which was most recently successfully applied by some of us to the ^71^Ga NMR parameters in reference [44], where further details can be found. In short, it applies the plane wave DFT-based calculations as implemented in the CASTEP 16.1 code [45] to first optimize the periodic structures of investigated crystals, and, subsequently, predict their NMR parameters by applying the GIPAW method [46,47]. In all calculations, the PBE functional [48] was used, and relativistic effects were treated by the ZORA method [49]. The CASTEP settings were consistent with the “Fine” accuracy level of the Materials Studio [50].

### 3.5. Program Description and Data Processing

UWNMRSpectralShape (USS) is a software with a graphical user interface (GUI) written in Java programming language (JDK 11.0) using an external JFreeChart library for plotting. The program was designed for fast and easy extraction of the final envelope of spikelets pattern of UWNMR spectra (for spectra to be processed properly they must be presented in magnitude (absolute value) mode; ‘mc’ command in TopSpin software) and its further transfer to conventional programs such as Bruker TopSpin and DMfit, or conversion into an ASCII file. The main interface of the software is shown in Figure 4. The GUI comprises three parts: (i) menu bar (top left), (ii) sidebar (left) and (iii) plot area (right). The menu bar contains options for conventional actions such as opening and saving files, showing the user guide, etc. The plot area displays the loaded NMR spectrum (Figure 4, black line). The sidebar is divided into sub-bars and contains radio-buttons for changing frequency units (ppm, Hz) and a text box indicating the basic transmitter frequency (O_1_ in MHz). The set of radio-buttons for switching between different processing options (after selecting an individual radio button, the corresponding option is enabled): (i) Finding spectral envelope, (ii) Smoothing mode, (iii) Background correction; buttons for Deleting spectral envelope, Cleaning up the plot area, Undo action, Zoom out and two buttons for Saving the final envelope in different formats.

Processing with USS software is intuitive and straightforward. After loading a UWNMR spikelets spectrum into the program, the sub-regions for creating spectral envelop need to be defined by a left mouse click in the plot area. These defined sub-regions and selected point(s) are then highlighted by vertical lines (see green lines in the inset magnification in Figure 4), and their positions are saved and used for subsequent processing. The coloring (green, magenta and gray) of the vertical lines depends on the processing option as follows:

*Defining spectral envelope (green; see manual):* Zoom in onto the spikelet signal(s) so that individual spikes are resolved. By clicking in between two consecutive spikes (as shown in Figure 4, where green lines are located), two sub-regions of the recorded NMR spectrum are defined. Then, the positions (on the x axis) of local maximum points in the defined sub-regions are automatically found using the “Linear search” algorithm:The first point of the region is set as maximum (*max*)FOR all points IN defined region DO (IF intensity of current point is higher than *max* THEN set current point as *max*)

As the distance between spikelets is constant (it depends only on experiment settings), the program automatically detects the local maximum of each spikelet. It builds the spectral envelope containing peak points of every spikelet and points from the noise region (which does not contain spikelets); see the red line in Figure 4. For several spectra, the results were checked against essentially the same procedure implemented in a MATLAB^®^script that is described in the Appendix A.

*Smoothing mode (magenta; see manual):* The generated spectral envelop (whole spectrum or defined region) can be smoothed by an implemented algorithm that averages the intensity of the local minimums relatively to the intensities of the neighboring points. This procedure slightly changes the intensity of the experimental data points in order to make the spectral profile look smoother; however, it has no impact on the obtained NMR parameters. This is performed using the following algorithm:FOR all points IN whole profile (or defined region) DO (IF intensity of the current point is lower than intensities of both (closest) neighboring points THEN intensity of the current point is equal to an average value of intensities of the neighboring points)

*Background correction (gray; see manual):* The noise region(s) of the recorded NMR spectra with a distorted baseline need to be defined using a left mouse click. After selecting “Make background correction”, the intensity of noise points in the specified region are set to an average value (±10%) of the intensity between the first and the last points in the selected region (see as an example Figure 3c, expt. and envelope). This procedure uses the following algorithm:An average intensity (*avg*) of the first and last points in the defined region is stored.FOR all points IN the defined region DO intensity is equal to *avg* ± 10%.

*Zoom:* The left mouse button can be used for zooming in. Alternatively, the mouse wheel can zoom the spectrum in and out. When a certain area of the spectrum is zoomed in, it is possible to drag and move the spectrum (along both axes) using a combination of “Ctrl” and left mouse button. The “Zoom out” option can be used to set the zoom to the default position.

A click of the right mouse button in the plot area leads to the appearance of a pop-up menu with plot options such as adding the title, editing axes and background, saving an image, etc. After processing the used UWNMR spectrum, the final continuous spectral line can be saved by clicking the “Save” or “Save as” button. The latter allows the user to save the file in a chosen folder, whereas the “Save” button will save results to the original folder (where the initial/opened file is located). Finally, the spectrum obtained and saved in this way can be used for automatic fitting of NMR parameters, such as quadrupolar coupling constant (*C_Q_*), asymmetry parameter (*η_Q_*) and chemical shift anisotropy (CSA), etc., in conventional NMR programs. More processing details are given in the program user guide: About>>How to use program. The latest program version is available at https://cutt.ly/evretIG (accessed on 29 September 2021).

## 4. Conclusions

A new approach for obtaining NMR parameters using the developed software was proven to be both faster and more straightforward than the method based on the time-domain co-addition of spin echoes. Additionally, it gave more reliable results in comparison to the conventional approach using a manual simulation and fitting of the powder spikelets pattern. It was shown that the software is eminently suitable for processing all spikelets NMR spectra yielded by quadrupolar and spin-^1^/_2_ nuclei and allows the successful analysis of broad (hundreds kHz), as well as relatively narrow (tens kHz), signals. Finally, the subsequent quantification of well-resolved, distinct species can be determined using the traditional integration of individual peaks. 

## Figures and Tables

**Figure 1 molecules-26-06051-f001:**
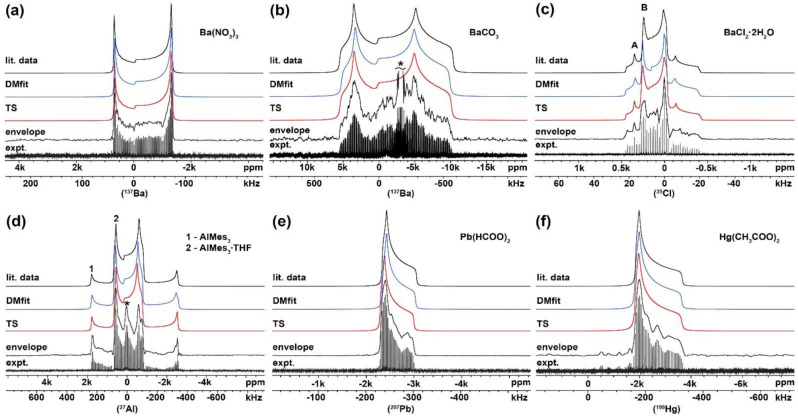
Experimental spikelets WURST-(Q)CPMG NMR spectra (expt.), envelope of the spikelets spectra defined by developed software (envelope), fitting obtained from Bruker TopSpin (TS) and DMfit (DMfit) software, and simulation based on literature data (lit. data) of: (**a**) (^137^Ba) Ba(NO_3_)_2_; (**b**) (^137^Ba) BaCO_3_; (**c**) (^35^Cl) BaCl_2_∙2H_2_O; (**d**, 1) (^27^Al) AlMes_3_; (**d**, 2) (^27^Al) AlMes_3_∙THF; (**e**) (^207^Pb) Pb(HCOO)_2_; (**f**) (^199^Hg) Hg(CH_3_COO)_2_. Impurities are denoted by asterisk (*).

**Figure 2 molecules-26-06051-f002:**
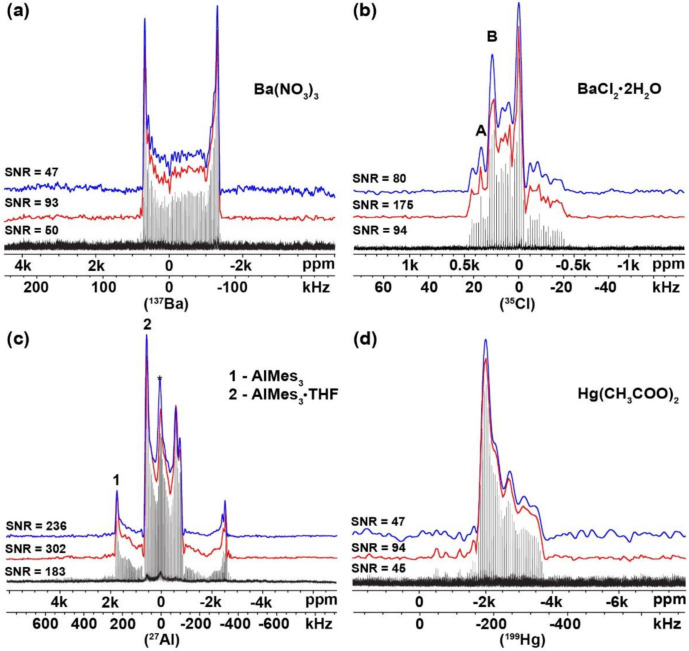
Experimental spikelets WURST-(Q)CPMG NMR spectra (black line), spectral profile of the powder pattern obtained by USS software (red line) and co-addition of spin echoes followed by Fourier transformation (blue line) of: (**a**) (^137^Ba) Ba(NO_3_)_2_; (**b**) (^35^Cl) BaCl_2_∙2H_2_O; (**c**, 1) (^27^Al) AlMes_3_; (**c**, 2) (^27^Al) AlMes_3_∙THF; (**d**) (^199^Hg) Hg(CH_3_COO)_2_.

**Figure 3 molecules-26-06051-f003:**
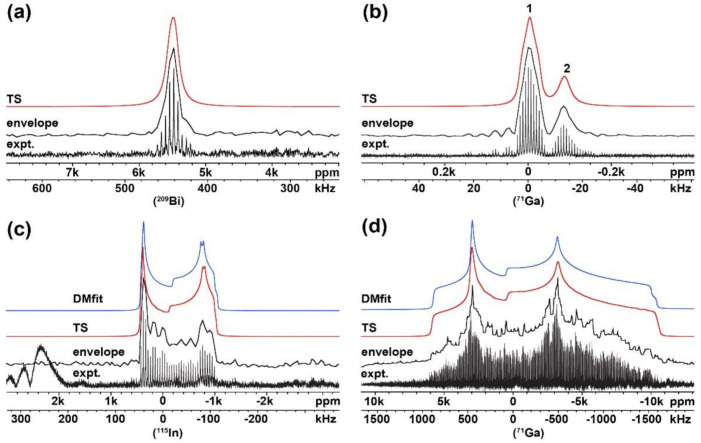
Experimental spikelets WURST-QCPMG NMR spectra (expt.), the envelope of the spectra defined by developed software (envelope), the fitting obtained from Bruker TopSpin (TS) software of: (**a**) (^209^Bi) Cs_2_AgBiBr_6_; (**b**) (^71^Ga) Ga (NO_3_)_3_∙9H_2_O; from Bruker TopSpin (TS) and DMfit (DMfit) software of: (**c**) (^115^In) In (NO_3_)_3_∙5H_2_O; (**d**) (^71^Ga) [Cl (Me) Ga (O^t^Bu)]_2_.

**Figure 4 molecules-26-06051-f004:**
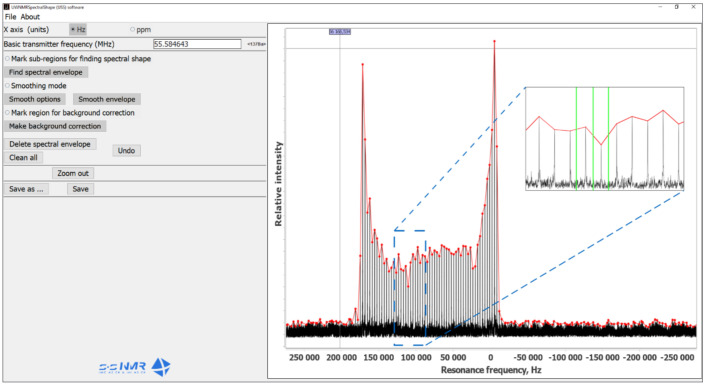
A screenshot of the main interface of UWNMRSpectralShape (USS) software.

**Table 2 molecules-26-06051-t002:** Comparison of NMR parameters obtained from the experiment and from the periodic DFT calculations.

Compound	Investigated Nuclei	*δ*_iso_ (ppm)	*σ*_iso_ (ppm)	*C*_Q_ (MHz)	*η* _Q_	*Ω* (ppm)	*κ*	*α* (deg)	*β* (deg)	*γ* (deg)	LB/GB (kHz)	
Cs_2_AgBiBr_6_ *	^207^Bi		4356	--- *	--- *	--- *	--- *	--- *	--- *	--- *	--- *	DFT
		5517 (25)		--- *	--- *	--- *	--- *	--- *	--- *	--- *	10.5/9.3	Expt. ^1^
Ga (NO_3_)_3_∙9H_2_O	^71^Ga (1)		1680	1.5	0.76	68	0.76	144	32	310	---	DFT
		−1 (5)		1.0 (0.5)	0.7 (0.10)	51 (10)	0.7 (0.2)	128 (40)	50 (20)	293 (20)	1.5/1.2	Expt. ^1^
Ga_13_(NO_3_)_15_(OH)_24_ ∙24H_2_O	^71^Ga (2)	−80 (10)		0.9 (0.4)	0.2 (0.1)	33 (10)	0.4 (0.2)	14 (20)	138 (30)	177 (25)	3/0.5	Expt. ^1^
In (NO_3_)_3_∙5H_2_O	^115^In		3745	50.4	0.19	314	0.61	90	20	270	---	DFT
		−80 (10)		45.7 (0.5)	0.14 (0.03)	117 (25)	0.3 (0.1)	60 (15)	10 (10)	300 (30)	1.4/0	Expt. ^1^
		−90 (10)		45.1 (0.5)	0.15 (0.03)	130 (25)	0.4 (0.1)	62 (15)	14 (10)	307 (30)	1.7/0	Expt. ^2^
[Cl (Me) Ga (O*^t^*Bu)]_2_	^71^Ga		1408	37.8	0.53	491	0.59	174	10	274	---	DFT
		286 (23)		44.3 (0.4)	0.49 (0.05)	546 (55)	0.7 (0.2)	135 (40)	12 (10)	273 (20)	16.2/17.1	Expt. ^1^
		307 (24)		43.7 (0.4)	0.49 (0.05)	543 (50)	0.6 (0.2)	130 (35)	18 (10)	288 (25)	15.6/0	Expt. ^2^

*δiso = (δ11 + δ22 + δ33)/3, CQ = eQV33/h, ηQ = (V11–V22)/V33, Ω = δ11–δ33, κ = 3(δ22-δiso)/Ω*; Euler angles are defined in [36]; ^1^ NMR parameters obtained using Bruker TopSpin software; ^2^ NMR parameters obtained using DMfit software. * The values are zero by symmetry.

## Data Availability

All data are included in the article.

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
