# Peer review of "Reconstructing Reliable Powder Patterns from Spikelets (Q)CPMG NMR Spectra: Simplification of UWNMR Crystallography Analysis"

_molecules, 2021, doi:10.3390/molecules26196051_

Round 1

Reviewer 1 Report

This is a very interesting but at the same time very technical manuscript.
To make it even more interesting for the average reader not particularly experienced in solid state NRM, it would be useful for the authors to implement the introduction by better explaining in which fields their research would have the greatest impact.   This can be done for example citing some application studies that show in detail the problems they intend to solve and discussing the benefits of applying their approach.
If this is not done, the work remains of interest only and exclusively to technicians specializing in the used NMR technique.

Author Response

We thank the reviewer for a very nice evaluation of our paper. We are aware that the topic is very specific for NMR users and not really aimed at readers uninvolved in the NMR field. However, we feel that the research is very important for the particular group of researchers and can greatly aide the understanding of this complicated analysis. We also feel that no further alterations to the Introduction will help the broader audience to get more interested in the topic if they do not already have some knowledge in ssNMR. Furthermore, the paper concept as well as introduction was praised by second and third reviewers.

Reviewer 2 Report

The authors developed software with GUI to automatically extract the ultra-broadband QCPMG spikelet spectra (envelope) into continuous spectra, which can be easily fitted in Bruker TopSpin solid-line shape tool or DMfit to estimate the anisotropic parameters. Several model compounds were investigated, and the extracted spectra were demonstrated to give fitting results either match the literature data or the DFT calculation. What's more, the extracted spectra provide a better signal-to-noise ratio than the direct accumulation of individual spin echo of the CPMG, which is an alternative approach to extra the envelope of the ultra-broadband spectra.

The motivation, experimental data, and subsequent analysis and comparisons were presented clearly. This report should be helpful for people dealing with ultra-broadband QCPMG spectra, which are ubiquitous in the field of materials solid-state NMR.

The only point worth raising is that showing all the NMR spectra in kHz unit instead of ppm unit might be easier for the readers, at least for me.

Author Response

We thank the reviewer for his/her praise of our work.

We have added kHz units in all Figures/spectra in MS as well as in SI as suggested by the reviewer.

Reviewer 3 Report

This paper presents a new software designed for users of NMR spectroscopy for the analysis of (Q)CPMG spectra. The principles of the (Q)CPMG method is well introduced and appropiate referencing to previous work is made (and I found in this respect the paper interesting and usefull for readers). The method proposed is not revolutionary but I think that the software proposed will indeed facilitate the quantitative analysis of such spectra (especially for new comers). In addition, nice experimental data are shown , some of them are new. I will therefore recommend to publish the paper as is. In addition, I will highly recomment the authors to add the orginal data (FID,) in the supp. mat for future users of the software and comparison with potential furture developments.

Typo : line 67 Hanh => Hahn.

Author Response

We thank the reviewer for her/his valuable comments. We have reread the manuscript and corrected typos.

Regarding the suggestion to add FID to the SI. There are several initial data files (including FID) included in the software under Examples folder, please download the software at https://cutt.ly/evretIG. We hope this is sufficient for showing how the program works.